# Stratification of Oligometastatic Prostate Cancer Patients by Liquid Biopsy: Clinical Insights from a Pilot Study

**DOI:** 10.3390/biomedicines10061321

**Published:** 2022-06-04

**Authors:** Antonella Colosini, Simona Bernardi, Chiara Foroni, Nadia Pasinetti, Andrea Emanuele Guerini, Domenico Russo, Roberto Bresciani, Cesare Tomasi, Stefano Maria Magrini, Lilia Bardoscia, Luca Triggiani

**Affiliations:** 1Department of Radiation Oncology, University and Spedali Civili Hospital, 25123 Brescia, Italy; a.colosini@unibs.it (A.C.); nadia.pasinetti@unibs.it (N.P.); a.guerini012@unibs.it (A.E.G.); stefano.magrini@unibs.it (S.M.M.); luca.triggiani@unibs.it (L.T.); 2CREA Laboratory (Centro di Ricerca Emato-Oncologica AIL), ASST Spedali Civili of Brescia, 25123 Brescia, Italy; simona.bernardi@unibs.it (S.B.); chiara.foroni@biologo.onb.it (C.F.); 3Unit of Blood Diseases and Bone Marrow Transplantation, Cell Therapies and Hematology Research Program, Department of Clinical and Experimental Sciences, University of Brescia, ASST Spedali Civili di Brescia, P.le Spedali Civili, 1, 25123 Brescia, Italy; domenico.russo@unibs.it; 4Radiation Oncology Service, ASST Valcamonica Esine, 25040 Esine, Italy; 5Division of Biotechnology, Department of Molecular and Translational Medicine (DMTM), University of Brescia, 25121 Brescia, Italy; roberto.bresciani@unibs.it; 6Department of Medical and Surgical Specialties, Radiological Sciences and Public Health, Section of Public Health and Human Sciences, University of Brescia, 25121 Brescia, Italy; cesare.tomasi@live.com; 7Radiation Oncology Unit, S. Luca Hospital, Healthcare Company Tuscany Nord Ovest, 55100 Lucca, Italy

**Keywords:** oligometastatic state, prostate cancer, stereotactic body radiotherapy, liquid biopsy, circulating cell free DNA, deep targeted sequencing

## Abstract

We propose a pilot, prospective, translational study with the aim of identifying possible molecular markers underlying metastatic prostate cancer (PC) evolution with the use of liquid biopsy. Twenty-eight castrate sensitive, oligometastatic PC patients undergoing bone and/or nodal stereotactic body radiotherapy (SBRT) were recruited. Peripheral blood samples were collected before the commencement of SBRT, then they were processed for circulating cell free DNA (cfDNA) extraction. Deep targeted sequencing was performed using a custom gene panel. The primary endpoint was to identify differences in the molecular contribution between the oligometastatic and polymetastatic evolution of PC to same-first oligo-recurrent disease presentation. Seventy-seven mutations were detected in 25/28 cfDNA samples: ATM in 14 (50%) cases, BRCA2 11 (39%), BRCA1 6 (21%), AR 13 (46%), ETV4, and ETV6 2 (7%). SBRT failure was associated with an increased risk of harboring the BRCA1 mutation (OR 10.5) (*p* = 0.043). The median cfDNA concentration was 24.02 ng/mL for ATM mutation carriers vs. 40.04 ng/mL for non-carriers (*p* = 0.039). Real-time molecular characterization of oligometastatic PC may allow for the identification of a true oligometastatic phenotype, with a stable disease over a long time being more likely to benefit from local, curative treatments or the achievement of long-term disease control. A prospective validation of our promising findings is desirable for a better understanding of the real impact of liquid biopsy in detecting tumor aggressiveness and clonal evolution.

## 1. Introduction

Oligometastatic prostate cancer (OPC) is a heterogeneous disease encompassing a broad spectrum of clinical states and related outcomes, such as de novo oligometastatic, oligorecurrent, and oligoprogressive disease [1,2,3,4]. The oligometastases theory was proposed for the first time in 1995 by Hellman and Weichselbaum, suggesting that metastatic dissemination occurs along an only apparent continuum from localized cancer to extensively metastatic disease [2,3]. Oligometastatic disease has been described to show a relatively limited metastatic potential [4]. In the natural history of prostate cancer, OPC may represent the initial step of an unavoidable, rapid progression to a polymetastatic state, or it may be the expression of a true oligometastatic phenotype; hence, a stable disease for a long time, and amenable to curative treatment or the achievement of long-term disease control [5,6,7]. To date, OPC is mainly defined according to a maximum number of distant metastatic lesions (usually 1 to 3) [8]. In the oligometastatic setting, the adoption of stereotactic body radiation therapy (SBRT) has proven to be a potentially curative treatment option. A large amount of retrospective, national, and international studies have already described SBRT as being safe and effective in terms of local control of disease and freedom from failure [9,10,11,12,13,14,15,16,17,18,19,20,21,22,23,24,25,26,27,28,29,30], owing to the postponement of the need for androgen deprivation therapy (ADT) prescription, which still represents the standard of care for metastatic prostate cancer. Such findings were recently confirmed by prospective series, and they definitely supported the role of metastases-directed therapy in OPC patients relapsing after a local treatment with a limited number of metastases [31,32,33,34].

Despite significant advances in understanding the clinical meaning of OPC, no biomarkers that differentiate between the oligometastatic and the polymetastatic state have been currently validated, and biological investigations for OPC patient stratification are still lacking in literature. Over the past years, molecular profiling has improved our knowledge regarding the genomic landscape of advanced prostate cancer [35,36]. For this purpose, the so-called “liquid biopsy” approach may provide a powerful tool for identifying predictive biomarkers and therapeutic targets in a non-invasive manner [37]. Moreover, molecular biology techniques such as next-generation sequencing (NGS) have emerged as a promising issue for characterizing the oligometastatic-state heterogeneity [38,39]. We present our pilot study, the translational analysis of serum-derived circulating cell free DNA (cfDNA) in a population of OPC patients, using a deep target sequencing approach, with the aim of tracking metastatic prostate cancer spread from a molecular point of view, the possible identification of biomarkers that are predictive for a true oligometastatic state, and we assess whether any molecular characterization of OPC may also contribute to select patients that are more likely to benefit from local, metastasis-directed SBRT [40].

## 2. Materials and Methods

### 2.1. Patients Selection and Treatment

The present study received final approval by the Institutional Ethical Committee, and it was performed in accordance with the principles of good clinical practice (GCP) with respect to the ICH GCP guidelines and the ethical principles contained in the Helsinki declaration [41].

The study population included adult patients with castrate-sensitive OPC, consecutively evaluated at our institution for SBRT on bone lesions (30 Gy/3 fractions, biological effective dose (BED) 108 Gy, considering α/ß 3 Gy) or nodal metastases (36 Gy/6 fractions, BED 100 Gy considering α/ß 1.5 Gy for cancer and 3 Gy for late-responder normal tissue). Inclusion criteria are reported in Table 1.

All of the metastatic lesions (bone and lymph nodes) had been detected by restaging 11C-Choline PET/CT scan, and/or 68Ga-Prostate-Specific Membrane Antigen (PSMA) PET/CT scan, at the time of biochemical failure after primary treatment with curative intent according to the European Association of Urology (EAU) guidelines [42].

Radiotherapy planning and treatment delivery are fully described in Appendix A—Radiotherapy Procedures.

Following SBRT, all the enrolled patients underwent clinical follow up as per usual, according to disease-specific, internationally accepted clinical practice [42] (details in Appendix A—Radiotherapy Procedures). For the purpose of the present protocol, data collection continued for a planned observation period of 36 months after the treatment of OPC.

### 2.2. Biological Experimental Plan

#### 2.2.1. cfDNA Extraction and Sequencing

Per each enrolled patient, peripheral blood samples were collected in 2 × 4.9 mL clot activator tubes (S-Monovette Sarstedt) before the start of SBRT, and kept at room temperature for 30–60 min to allow for clotting, and then centrifuged at 2000× *g* for 15 at 4 °C. Circulating cell free DNA (cfDNA) was purified from serum using QIAamp Circulating Nucleic Acid kit (Qiagen, Hilden, Germany) according to the manufacturer’s instructions. cfDNA was quantified using QuantiT™ Oligreen^®^ ssDNA kit (Thermofisher, Waltham, MA, USA) and Infinite200 plate reader (Tecan, Männedorf, Switzerland).

#### 2.2.2. Library Preparation and NGS Analysis

After extraction, cfDNA from serum immediately underwent end-repair, A-tailing, and ligation to Illumina indexed adapters.

A custom-targeted gene panel was designed following the evidences reported in the literature regarding the molecular characterization of prostate cancer [35,43], using the NimbleDesign tool (Roche, Basel, Switzerland), and included the coding sequence (exons) of the genes showed in Table 2.

Once the DNA libraries were indexed, they were PCR-amplified, quantified, and pooled before hybridization to a custom NimbleGen SeqCap EZ Choice Library (Roche) of oligonucleotide DNA probes that were complementary to the coding sequences of the 37 selected genes (Table 2). Regulatory sequences and splicing sites were also included. The pooling strategy was 8 samples per pool. The hybridization was performed over-night for 18 h. After stringent washing, the captured libraries were PCR-amplified and sequenced to generate 2 × 150 bp paired-end reads with a MiSeq platform (Illumina, San Diego, CA, USA). Finally, the resulting DNA sequences were automatically demultiplexed and aligned to the human reference genome Hg19. Sequence variants were detected and annotated by Webannovar.

The thresholds for the candidate variants were 1000× for locus coverage, and 50 total reads (25 reads on both sequenced DNA strands) reporting the variant [44].

Common polymorphisms (≥1% in the general population) were discarded by comparison with NCBI, dbSNP, 1000 genomes, and EXAC, and then automatically investigated by Webannovar. However, since these databases contain known disease-associated mutations, all detected variants were compared with the gene-specific mutation databases, ClinVar and COSMIC. Then, we screened for mutations that could give rise to premature protein-truncating mutations or those with a high impact on the protein structure, such as, stop mutations, missense variants, mutations on splice sites, and exonic indels.

Unknown variants were automatically ranked by Webannovar pipeline. This ranking was based upon the evolutionary conservation and potential level of danger of the affected nucleotide locus, using Sift, Polyphen2, Mutation tester, FATHMM, ProVean, MetaSVM, and M-CAP [44].

Mutation rates (per megabase) was determined dividing the mutation count for each sample by the extension gene panel (0.1092 Mb).

### 2.3. Statistical Analysis

The database was formatted using Microsoft Excel^®^ software and later imported from IBM-SPSS^®^ software ver. 26.0.1 (IBM SPSS Inc. Chicago, IL, USA). The use of the Stata^®^ software ver. 16.0 (Stata Corporation, College Station, TX, USA) was also considered for comparisons or implementations of the test output.

#### Normality of the Distributions as Assessed Using the Kolmogorov–Smirnov Test

Categorical variables were presented as frequencies or percentages, and compared with the use of the Chi-Square test and the Fisher’s exact test, as appropriate; associations of the crosstabs were verified using standardized adjusted residuals.

Continuous variables were presented as means ± standard deviation (SD) (in the case of a normal distribution), or medians, and min/max (in the case of a skewed distribution) and compared with the use of Student’s T-test, ANOVA, or the Mann–Whitney and Kruskal–Wallis test; correlations among variables were analyzed by the Pearson’s or Spearman’s rank correlation test. A stepwise logistic regression analysis including all variables with probability values < 0.05 in the univariate analysis was used to determine independent predictors; ATM no/yes was the dependent variable, and overall ADT treatment time, FOXA1 yes/no, site of recurrence after primary treatment (nodal, bone or both), local control disease yes/no, BRCA1 yes/no, POLD1 yes/no, were the independent variables. The results are presented as an odds ratio (OR) with 95% confidence intervals. A two-sided α level of 0.05 was used for all tests. The authors had full access to and take full responsibility for the integrity of the data.

## 3. Results

### 3.1. Population Clinical Characteristics and Outcomes

A total of 28 patients with castrate sensitive, oligorecurrent prostate cancer were considered suitable for this study between November 2017 and July 2018 at the Spedali Civili Hospital of Brescia. The median age was 67 years (52–75). Population characteristics and treatment outcomes are summarized in Table 3.

All of the treated lesions were detected by 11C-Choline PET-TC, then used to support radiation treatment planning. Nodal metastases were recorded in 20 (71.5%) patients, while 6 (21.4%) of them had bone lesions; both nodal and bone involvement were reported in 2 (7.1%) cases.

At the end of the observation period, a median follow-up of 34.9 months (range 17.4–43.7) was reported. We recorded an 83% local control of disease after the first SBRT; grade 1 acute bladder toxicity in 1 case; grade 2 late bowel toxicity; no grade ≥ 3 acute or late toxicity. The median ADT-free survival was 19.5 months (range 5.5–43.9).

### 3.2. Genomic Landscape of Oligometastatic Prostate Cancer

Blood samples were collected from all patients before radiation treatment and approximately 2.5 mL of serum were processed for cfDNA. The median [cfDNA] was 30.2 ng/mL (range 4.2–171.6 ng/mL) (Figure 1).

Deep targeted DNA sequencing of 37 prostate cancer relevant genes was performed, and a total of 77 mutations were detected in 25 of 28 cfDNA samples (Figure 2). A detailed list of the detected mutations is reported in Appendix B, Table A2.

The median tumor mutational load was 27.5/Mb (range, 0–73.3 mutations/Mb) (Figure 3).

Genomic alterations mainly occurred in the AR and DNA repair gene pathways. Nine different deletions, two insertions, and one missense mutation regarding AR genes were identified in 13 (46.4%) patients. Among these, four (30.8%) patients carried the 171_176 deletions and two (15.4%) others, the 171_182 deletion. Furthermore, alterations in the FOXA1 gene were detected (3 (10.7%) cases). ATM mutations were identified in 14 (50.0%) patients in the cohort, and most of them were missense mutations. Intriguingly, six (42.9%) patients harbored the 5557 G > A mutation, two (14.3%) of them carried the 3161 G > C mutation, one (7.1%) patient had a stopgain mutation, and one (7.1%) a frameshift deletion. We observed frameshift deletions and missense mutations in BRCA2 in 11 (39.3%) patients, eight of whom (72.7%) harbored the 1114 C > A aberration. Patient 08 carried a frameshift deletion in the BRCA2 gene, causing a truncated protein product. Six (21.4%) patients showed missense mutations in th BRCA1 sequence, and three (50.0%) of them harbored the 1936 A > G aberration. POLE mutations were identified in five (17.9%) patients, four (80.0%) of them showing the 755 C > T mutation. Four (14.3%) patients carried a 158 G > A alteration in the RAD51D gene, and one (3.6%) patient harbored a missense mutation in MSH2. A loss of heterozygosis was also observed: two patients harboring the 1114 C > A BRCA2 alteration, one patient carrying the 1936A > G BRCA1 mutation, and one patient with the 158A > G RAD51D variant. Missense mutations in HGF were also observed (three (10.7%) cases). Four (14.3%) patients carried ETS gene fusion variants: two (7.1%) patients with missense mutations in ETV4, and two (7.1%) patients with a missense mutation and a stopgain mutation, respectively. Finally, missense mutations in the IDH1 (one (3.6%) patient), ALDH1A1 (one (3.6%)), and SPARC (one (3.6%)) stemness genes were identified. We did not observe mutations in the PI3KCA pathway, nor in tumor suppressor genes. 

When analyzing the interaction between the identified genomic alterations and pre-treatment [cfDNA], we found a lower median [cfDNA] in ETV4 carriers than in ETV4 non-carriers (6.6 ng/mL and 33.35 ng/mL, respectively) (*p* = 0.021), together with a lower median [cfDNA] in ATM carriers than ATM non-carriers (24.02 ng/mL and 40.035 ng/mL, respectively) (*p* = 0.039).

The interaction between the identified genomic alterations and clinical parameters was also investigated. There was a trend for an increased risk for ADT prescription requirement in ATM mutation carriers (OR = 0.160, *p* = 0.057), while SBRT failure was associated with an increased risk of harboring BRCA1 mutations (OR = 10.5, *p* = 0.043).

## 4. Discussion

Metastatic prostate cancer has been demonstrated to be molecularly heterogeneous, with clinical heterogeneous behavior [45,46]. To date, many treatment options are available for the management of metastatic prostate cancer; nevertheless, there are some unmet needs for patient stratification in this setting [47]. In this scenario, biomarker discovery using liquid biopsy has become an interesting field of study [37,48]. There is a growing interest for deep targeted cfDNA sequencing as a promising way of identifying evolution markers underlying progressive cancer or the metastases process before clinical onset [37]. DNA sequencing approaches based on next-generation (NGS) technology have enabled the rapid, high-quality analysis of genes that may be relevant for disease-specific characterization. NGS can generate wide and complete masses of DNA sequence data, and perform quick and cost-effective genetic analysis [38]. Currently, there is only one Food and Drug Administration (FDA)-approved liquid biopsy assay for the detection of BRCA1, BRCA2, and/or ATM aberrations in patients with metastatic castrate-resistant prostate cancer (mCRPC) who may be appropriate for the treatment with poly(ADP-ribose) polymerase (PARP) inhibitors (PARPi) [49,50]. Compared to tissue biopsy, liquid biopsy is non-invasive and captures aggregate features from tumor material-releasing metastases, thus providing real-time information on the state of the disease. It may be a useful tool for treatment resistance tracking [37,48,51].

OPC represents a phenotype with limited metastatic potential, and reveals heterogeneity in clinical behavior [1,4]. Metastases-directed, ablative SBRT has been gaining validation as a safe and effective treatment option for oligorecurrent/oligoprogressive PC [52]. Beyond the large number of promising findings from retrospective data, the prospective, phase II Observation versus stereotactic ablative RadiatIon for OLigometastatic Prostate CancEr Trial (ORIOLE) recently confirmed SBRT to improve oncologic outcomes in a certain setting of oligometastatic PC patients, with an excellent local control of disease and mild adverse events, with no impact on the patient quality of life [33]. Preliminary results from the randomized, phase II Surveillance or metastasis-directed Therapy for OligoMetastatic Prostate cancer recurrence (STOMP) trial also supported the utility of metastasis-directed therapy (surgery or SBRT) in the oligometastatic state of PC [31,32].

Despite the excellent local control and improved survival obtained with such a metastasis-directed treatment approach (i.e., surgery or ablative SBRT), sometimes OPC only represents the initial step of a rapid, unavoidable progression to a polymetastatic disease [7]. The identification of features that discriminate a true oligometastatic state and a polymetastatic progression phenotype is challenging. Since the nucleic acids released into the blood stream are considered to be cell messengers, information regarding a true oligometastatic condition and/or tumor cell reactions to radiation may be derived [53]. However, unlike colorectal or lung cancer, there are no recurrent point mutations in PC that can be used to track the disease. Moreover, relevant to intra-patient heterogeneous diseases, different sites of metastases may likely have different DNA release rates, and the ability to detect cfDNA-based features is gene region- and individual dependent [54,55,56,57].

We planned a prospective, pilot study since there are no similar biological study designs to compare with for accrual evaluation that have ever been reported in the literature.

The primary endpoints were differences in molecular contribution between the oligometastatic and polymetastatic evolution of prostate cancer to same-first oligorecurrent disease presentation, and their association with ADT-free survival (ADT-FS), defined as the time between the first day of SBRT and the start of palliative ADT. The secondary endpoints were: distant progression-free survival (DPFS, defined as the time between the first day of SBRT and the detection of clinical disease outside the PTV after further biochemical progression); local control (in-field control) of disease (defined as no evidence of disease in the treatment field at the restaging Choline- or PSMA-PET/CT).

Our exploratory analysis revealed that cfDNA concentrations in patients with OPC were relatively low, considering those reported in literature for patients with mCRPC, which is a more advanced and prognostically unfavorable stage of the disease [58]. Despite the absence of a direct comparison between the two groups in our study, nor in the available literature, taking into account the current biological evidence regarding cancer metastatization [2,3], we may hypothesize that such lower cfDNA concentration is likely to reflect the limited metastatic potential of the oligometastatic state [58].

The distribution of the genomic alterations in our study were highly consistent with the genomic landscape of prostate cancer described in literature [45,46]. As expected, mutations in the AR and DNA repair genes included in our panel were the most common ones, in particular, the ATM and BRCA1/BRCA2 genes, and they seemed to drive high tumor mutation load and rapid polymetastatic spread after the first oligorecurrence treated with SBRT. In detail, most of the AR aberrations consisted of non-frameshift deletions, and they were located in a possible critical and hotspot region of the N-terminal domain of the receptor. We did not observe mutations in the ligand-binding domain, as is commonly mapped [59]. BRCA1/2 and ATM aberrations have been reported to be associated with a more aggressive prostate cancer phenotype, with an increased risk of disease recurrence and poorer survival outcomes [60,61,62]. In our cohort, the ATM 3161 G > C variant was associated with an increased prostate cancer risk [63]. Our findings also showed that the BRCA2 mutation rate was higher than BRCA1, in line with the available literature. BRCA2 mutations are known to be associated with a higher prostate cancer mortality, likely due to the direct involvement of BRCA2 in the homologous recombination process, as it mediates the recruitment of RAD51 to DNA double-strand breaks [61]. Of note, although not clearly statistically significant, patients in our series with a history of ADT prescription before metastatic onset (concurrent to primary or salvage treatment) were more likely to have an ATM mutation. However, there are not enough elements to establish whether these are germinal mutations probably harboring a poorer prognosis, or somatic alterations induced by ADT itself. Not less important, OPC patients with combined mismatch repair gene and BRCA2 mutations seemed to have a worse prognosis. Nearly half of BRCA1-mutated OPC patients also showed no local control of the disease after the first SBRT course. All this being considered, our preliminary data suggested that the presence of high-risk prostate cancer mutations (regarding ATM, BRCA1, and BRCA2 genes) might be the starting point for identifying a subset of patients who may benefit from systemic treatment and local radiotherapy in combination strategies since the first clinical and radiological evidence of metastatic, castrate-sensitive PC.

In the near future, the PC oligometastatic state should be better characterized based on its molecular features, not only considering the number of metastatic lesions [1,6,64].

To our knowledge, our series may be considered the first prospective one with translational implications in the field of oligometastatic PC.

Unfortunately, the relatively small sample size of our study is the main limitation of our study, and it did not allow much powerful conclusions to be drawn. However, our prospective pilot study provided some promising clinical insights regarding the possible identification of metastatic prostate cancer evolution markers. Whether our results were confirmed for a wider cohort of patients, it would be possible to set up gene clustering for a highly accurate prognostic stratification of patients with metastatic PC.

Tracking molecular evolution markers predictive for a better response to stereotactic irradiation, or for a worse prognosis, which supports the need for systemic therapy, may offer some clues for defining the best radiation treatment features and technological solutions in the rapidly expanding field of stereotactic cancer treatments. Of note, the identification of a true oligometastatic state that is amenable to local, ablative treatment may also allow for the postponement of the need for ADT in a significant number of patients, ensure better patient QoL, a reduction in possible side effects (i.e., metabolic syndrome, climacteric syndrome, etc.) and reduced healthcare costs.

## 5. Conclusions

We proposed a preliminary, exploratory analysis of serum-derived cfDNA samples collected before SBRT treatment through a deep target sequencing approach. We investigated whether the sequencing of 37 prostate cancer-relevant genes might contribute to the recognition of the heterogeneity of oligometastatic prostate disease, and a better molecular characterization of oligometastatic disease. Moreover, we attempted to examine the obtained molecular data, to predict an eventual polymetastatic progression.

Our study may represent a crucial foundation for the future design of clinical trials owing to provide the stratification of patients with OPC based on a molecular signature. Moreover, the assessment of a molecular fingerprint for OPC may allow for the identification of patients with a true oligometastatic phenotype, and hence, with stable disease for a long time, which are more likely to benefit from local, curative treatments, or the achievement of long-term disease control. The prospective validation of our promising findings on a wider series is desirable for a better understanding of the real impact of liquid biopsy analysis in detecting tumor features. Whether our findings were confirmed on a large scale, liquid biopsy might become a useful source of prognostic and predictive markers of metastatic prostate cancer spread. The real-time molecular characterization of cancer disease by liquid biopsy is also expected to be crucial for setting a patient-tailored therapeutic approach based on tumor aggressiveness and clonal evolution, to be applied a long time before clinical onset and the subsequent clinical appearance of advanced and extended progression of disease.

## Figures and Tables

**Figure 1 biomedicines-10-01321-f001:**
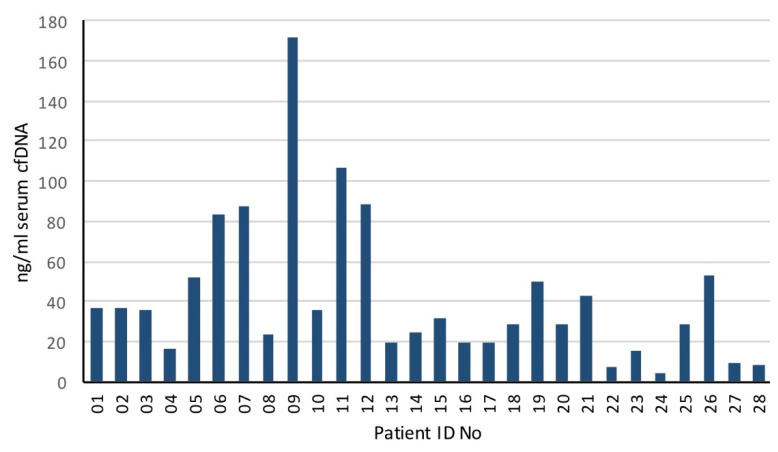
Serum cfDNA concentration (ng/mL) in oligometastatic prostate cancer patients.

**Figure 2 biomedicines-10-01321-f002:**
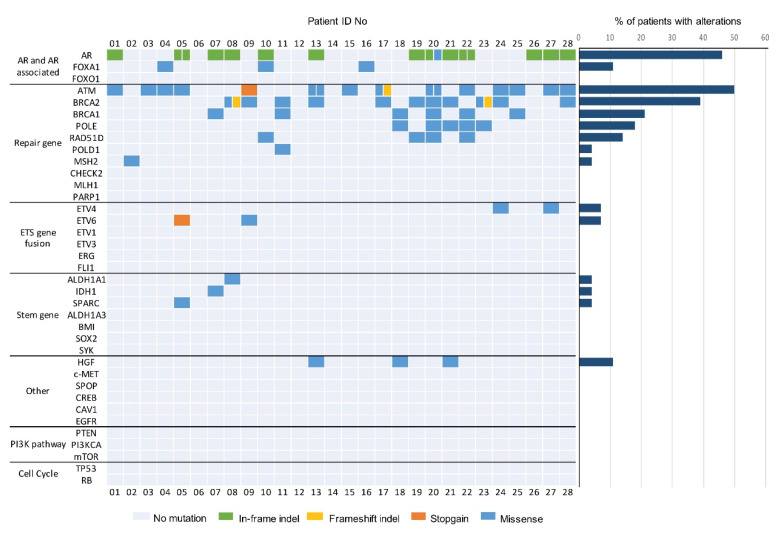
Genomic landscape of oligometastatic prostate cancer from targeted serum cfDNA sequencing. Oncoprint shows genomic alterations identified in cfDNA of patients with oligometastatic prostate cancer. Genes are grouped by pathway (37 genes shown). Mutational frequency for each gene in the targeted panel is provided on the right.

**Figure 3 biomedicines-10-01321-f003:**
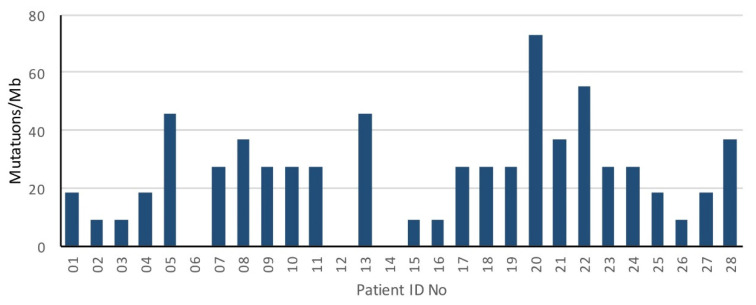
Mutation rate of oligometastatic prostate cancer patients.

**Table 1 biomedicines-10-01321-t001:** Patient selection criteria.

**Inclusion Criteria**
18 years old Pathologically confirmed acinar adenocarcinoma of prostate Castrate-sensitive OPC: ≤ 3 lesions (bone or node) detected with choline/PSMA PET following prostate specific antigen (PSA) rising after primary treatment with curative intent as defined by European Association of Urology criteria (EAU) Patients eligible for a course of SBRT Patients amenable to sign written informed consent
**Exclusion criteria**
Ongoing ADT (stopped <6 months before baseline evaluation) Prior treatment for castrate-sensitive OPC Testosterone levels <50 ng/mL

**Table 2 biomedicines-10-01321-t002:** Genetic panel for bioinformatics analysis.

TP53	PIK3CA	mTOR	FOXA1	FOXO1	BRCA1	BRCA2
PTEN	CREB	AR	ETV1	ETV3	ETV4	ETV6
ALDH1A1	ALDH3A1	SPOP	FLI1	IDH1	ERG	SOX2
cMET (HGFR)	HGF	SPARC	CAV1	BMI1	PARP1	RB1
ATM	CHEK2	EGFR	POLE	POLD1	MSH2	MLH1
RAD51D	SYK					

**Table 3 biomedicines-10-01321-t003:** Population characteristics.

**Age**	N	%	**Residual Disease after Surgery**		N	%
<65 years	13	46.4	R0 (no)		8	28.6
≥65 years	15	53.6	R1 (microscopic)		15	53.6
			R2 (macroscopic)		0	0.0
**T stage**	N	%				
T1c	3	10.7	**EBRT ^1^ (total dose)**		N	%
T2c	6	21.4	66 Gy		6	21.4
T3a	11	39.4	70 Gy		18	64.2
T3b	6	21.4	74 Gy		1	3.6
T4	2	7.1				
			Elective node irradiation		2	7.1
**N stage**	N	%				
N0	24	85.7	**Adjuvant ADT ^1^**		N	%
N1	4	14.3	No		21	75.0
			LHRH-analogue		2	7.1
**Gleason Grade Group**	N	%	Antiandrogen		4	14.3
1	6	21.4	Total Androgen Blockade		1	3.6
2	11	39.4				
3	3	10.7	**Biochemical relapse after primary treatment**		N	%
4	2	7.1	Yes		27	96.4
5	6	21.4	No		1	3.6
**D’Amico Risk Class**	N	%	**Biochemical control duration**		N	%
Very low	0	0,0	<1 year		5	17.9
Low	2	7.1	1–5 years		15	53.6
Favorable intermediate	4	14.3	>5 years		8	28.5
Unfavorable intermediate	2	7.1	Median bRFS^1^ 42.4mo (range 1.9–133.1)			
High	12	42.9				
Very high	8	28.6	**ADT ^1^ for biochemical relapse**		N	%
			No relapse		1	3.6
**Primary treatment**	N	%	No		17	60.7
Surgery	1	3.6	LHRH-analogue		6	21.4
EBRT ^1^	3	10.7	Antiandrogen		3	10.7
Brachytherapy (LDR 145 Gy)	2	7.1	Total Androgen Blockade		1	3.6
Surgery+Adjuvant RT ^1^	9	32.2				
Surgery+Salvage RT ^1^	13	46.4	**Number of treated lymph nodes**		N	%
			1 lymph node		14	50.0
**Oligorecurrence site**	N	%	2 lymph nodes		5	17.8
Nodal	20	71.5	3 lymph nodes		2	7.1
Bone	6	21.4	4 lymph nodes		1	3.6
Both	2	7.1				
			**Bone SBRT ^1^ target**		N	%
**Nodal SBRT ^1^ target**	N	%	Axial		4	14.3
Pelvic lymph nodes	19	67.8	Extra-axial		4	14.3
Abdominal lymph nodes	2	7.1	Hip		3	10.7
Both	1	3.6	Sternum/Ribs		2	7.1
One nodal region	18	64.2	One site		6	21.4
More than one nodal region	4	14.3	Two sites		2	7.1

^1^ EBRT = External-Beam Radiation Therapy; LDR = Low Dose Rate; RT = Radiation Therapy; SBRT = Stereotactic Body. Radiation Therapy; ADT = Androgen Deprivation Therapy; bRFS = biochemical Relapse-Free Survival.

## Data Availability

The data presented in this study are available on request from the corresponding author. The data are not publicly available due to Institutional policies, and the need for the authorization of the Study Coordinator (L.T. and S.M.M.).

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
