# Peer review of "Stratification of Oligometastatic Prostate Cancer Patients by Liquid Biopsy: Clinical Insights from a Pilot Study"

_biomedicines, 2022, doi:10.3390/biomedicines10061321_

Round 1
Reviewer 1 Report
The study by Colosini and coworkers reports a pilot analysis of the sequencing of 37 genes (relevant in prostate cancer - PCa) from 28 patients with an oligometastatic PCa before SBRT (stereotactic body radiotherapy). The main aim of the study is to identify mutations in these genes from liquid biopsies to predict the response to SBRT.
Overall, I found the introduction of the study confusing as I first thought that the aim was to identify markers discriminating oligometastatic from polymetastatic PCa. The introduction must be rewritten to better fit with the content of the study and provide additional background as many studies were performed to predict and evaluate SBRT response in PCa.
I also found it confusing that :
1) many conclusions are based from the comparison of data gathered by the group vs the rest of the litterature. For instance, « cfDNA concentration in patients with OPC were lower than those reported in patients with mCRPC ». This is not tested in the manuscript and protocols as well as platforms to prepare cDNAs might be quite different between studies. A comparison is not possible. This must be presented differently.
2) some conclusions are not significant : « ATM carriers had an increased risk to need for ADT prescription (p=0.057) ». The conclusion is not true and must be removed.
3) some conclusions are bias because of the design of the study. « Genomic alterations mainly occured in AR and DNA repair gene pathway ». Since 1/3 of the genes tested are in this class it is somehow expected to find a significant number of genes in these categories. Needless to say that the genes are also frequently mutated in other cancers and associated with metastasis. This conclusion must be tune-down.
In conclusion, I found the principle of the study very interesting, and some limitations are clearly stated at the end of the document. I would be happy to recommend publication if the aim and conclusions of the study were in line with the data.
Author Response
The study by Colosini and coworkers reports a pilot analysis of the sequencing of 37 genes (relevant in prostate cancer - PCa) from 28 patients with an oligometastatic PCa before SBRT (stereotactic body radiotherapy). The main aim of the study is to identify mutations in these genes from liquid biopsies to predict the response to SBRT.
Overall, I found the introduction of the study confusing as I first thought that the aim was to identify markers discriminating oligometastatic from polymetastatic PCa. The introduction must be rewritten to better fit with the content of the study and provide additional background as many studies were performed to predict and evaluate SBRT response in PCa.
We apologize for the inconvenience.
We chose differences in molecular contribution between oligometastatic and polymetastatic evolution of prostate cancer to same-first oligorecurrent disease presentation as primary endpoint of our explorative study.
We aimed to track metastatic prostate cancer spread from a molecular point of view, check if specific biomarkers predictive for a true oligometastatic state may exist, and to assess whether any molecular characterization of oligometastatic prostate cancer may also contribute to select patients more likely to benefit from local, metastasis-directed SBRT than those with a rapid polymetastatic evolution, hence needing for upfront systemic therapy.
The introduction was reviewed to make the aims of our study clearer.
Additional background on the role of SBRT in PCa was also provided.
I also found it confusing that :
1) many conclusions are based from the comparison of data gathered by the group vs the rest of the litterature. For instance, « cfDNA concentration in patients with OPC were lower than those reported in patients with mCRPC ». This is not tested in the manuscript and protocols as well as platforms to prepare cDNAs might be quite different between studies. A comparison is not possible. This must be presented differently.
We apologize for the inconvenience.
We are aware that we did not perform a direct comparison between patients with “Oligometastatic prostate cancer” and “mCRPC”, either such cohorts of patients cannot directly be compared since they represent different stages of disease for clinical characteristics, evolution, and prognosis.
We just tried to speculate on the relatively low cfDNA concentration in patients with evidence of a true oligometastatic, indolent evolution of disease in our explorative study, and made some hypoteses on how to exploit such findings to improve knowledge about OPC in relation to our study endpoints.
This section of the discussion has been reviewed and updated.
2) some conclusions are not significant : « ATM carriers had an increased risk to need for ADT prescription (p=0.057) ». The conclusion is not true and must be removed.
We apologize for the inconvenience.
The p=0.057 represents a trend for increased risk to need for ADT prescription in ATM mutation carriers. We believe that such findings, although not clearly statistically significant, cannot be ignored in view of the current evidence about the genomic landscape of cancer tumorigenesis, and related therapeutic advances in the setting of prostate cancer, but also other type of tumors.
3) some conclusions are bias because of the design of the study. « Genomic alterations mainly occured in AR and DNA repair gene pathway ». Since 1/3 of the genes tested are in this class it is somehow expected to find a significant number of genes in these categories. Needless to say that the genes are also frequently mutated in other cancers and associated with metastasis. This conclusion must be tune-down.
Thank you for your suggestion.
This section of the discussion has been reviewed and updated.
SInce it was an exploratory study, we chose to select a custom-targeted gene panel following the evidences reported in literature about molecular characterization of prostate cancer
In conclusion, I found the principle of the study very interesting, and some limitations are clearly stated at the end of the document. I would be happy to recommend publication if the aim and conclusions of the study were in line with the data.
Thank you for your appreciation and suggestions.
Reviewer 2 Report
The authors explore the expression of different mutations in the DNA profile using liquid biopsy in patients with oligometastatic disease. This is a mere exploratory study.
One of the pitfalls is that a comparison between the expression of mutation in cfDNA is not performed along that of the liquid biopsy, the gold standard.
Are the mutations identical and can these mutations observed in the liquid biopsies be used for predictive studies?
Author Response
The authors explore the expression of different mutations in the DNA profile using liquid biopsy in patients with oligometastatic disease. This is a mere exploratory study.
One of the pitfalls is that a comparison between the expression of mutation in cfDNA is not performed along that of the liquid biopsy, the gold standard.
Are the mutations identical and can these mutations observed in the liquid biopsies be used for predictive studies?
Thank you for your appreciation and suggestions.
We added a little integration to the conclusions.
Round 2
Reviewer 1 Report
I thank the authors for taking into account my comments and I have no further request.
Reviewer 2 Report
The manuscript has been improved by the authors and the pitfalls have been addressed.